# The Potential for High-Priority Care Based on Pain Through Facial Expression Detection with Patients Experiencing Chest Pain

**DOI:** 10.3390/diagnostics15010017

**Published:** 2024-12-25

**Authors:** Hsiang Kao, Rita Wiryasaputra, Yo-Yun Liao, Yu-Tse Tsan, Wei-Min Chu, Yi-Hsuan Chen, Tzu-Chieh Lin, Chao-Tung Yang

**Affiliations:** 1Department of Emergency Medicine, Taichung Veterans General Hospital, Taichung 407219, Taiwan; 2Department of Industrial Engineering and Enterprise Information, Tunghai University, Taichung 407224, Taiwan; rita.wiryasaputra@ukrida.ac.id; 3Informatics Department, Krida Wacana University, Jakarta 11470, Indonesia; 4Department of Computer Science, Tunghai University, Taichung 407224, Taiwan; 5School of Medicine, Chung Shan Medical University, Taichung 402306, Taiwan; 6Department of Emergency Medicine, Institute of Occupational Medicine, Taichung Veterans General Hospital, Taichung 407219, Taiwan; 7Department of Occupational Safety and Health Office, Taichung Veterans General Hospital, Taichung 407219, Taiwan; 8Department of Post-Baccalaureate Medicine, College of Medicine, National Chung Hsing University, Taichung 407224, Taiwan; williamchu0110@gmail.com; 9School of Medicine, National Yang Ming Chiao Tung University, Taipei 112304, Taiwan; 10Department of Family Medicine, Taichung Veterans General Hospital, Taichung 407219, Taiwan; 11Geriatrics and Gerontology Research Center, College of Medicine, National Chung Hsing University, Taichung 407224, Taiwan; 12College of Fine Arts and Creative Design, Tunghai University, Taichung 407224, Taiwan; 13Research Center for Smart Sustainable Circular Economy, Tunghai University, Taichung 407224, Taiwan

**Keywords:** cardiovascular disease, chest pain, deep learning, facial expression, expression recognition, YOLO

## Abstract

Background and Objective: Cardiovascular disease (CVD), one of the chronic non-communicable diseases (NCDs), is defined as a cardiac and vascular disorder that includes coronary heart disease, heart failure, peripheral arterial disease, cerebrovascular disease (stroke), congenital heart disease, rheumatic heart disease, and elevated blood pressure (hypertension). Having CVD increases the mortality rate. Emotional stress, an indirect indicator associated with CVD, can often manifest through facial expressions. Chest pain or chest discomfort is one of the symptoms of a heart attack. The golden hour of chest pain influences the occurrence of brain cell death; thus, saving people with chest discomfort during observation is a crucial and urgent issue. Moreover, a limited number of emergency care (ER) medical personnel serve unscheduled outpatients. In this study, a computer-based automatic chest pain detection assistance system is developed using facial expressions to improve patient care services and minimize heart damage. Methods: The You Only Look Once (YOLO) model, as a deep learning method, detects and recognizes the position of an object simultaneously. A series of YOLO models were employed for pain detection through facial expression. Results: The YOLOv4 and YOLOv6 performed better than YOLOv7 in facial expression detection with patients experiencing chest pain. The accuracy of YOLOv4 and YOLOv6 achieved 80–100%. Even though there are similarities in attaining the accuracy values, the training time for YOLOv6 is faster than YOLOv4. Conclusion: By performing this task, a physician can prioritize the best treatment plan, reduce the extent of cardiac damage in patients, and improve the effectiveness of the golden treatment time.

## 1. Introduction

Cardiovascular disease (CVD), one of the chronic non-communicable diseases (NCDs), is defined as a cardiac and vascular disorder that includes coronary heart disease, heart failure, peripheral arterial disease, cerebrovascular disease (stroke), congenital heart disease, rheumatic heart disease, and elevated blood pressure (hypertension). CVD is the leading cause of death globally, causing an estimated 18.6 million deaths annually [1,2]. Moreover, climate change, lifestyle, and genetics have increased CVD and made it a significant public health concern [3]. According to the World Health Organization (WHO), out of five CVD deaths, over 80% are due to heart attacks—as one of the coronary heart diseases—and strokes, and one-third of these deaths occur to people below the age of 70 [4]. The golden hour of a heart attack is a very critical and requires immediate medical attention. Chest pain or chest discomfort is one of the symptoms of a heart attack. The impact of a delay in first-degree emergency care lasting four to six minutes is brain cell death. Although the patient can be saved within the first ten minutes, the patient may become unconscious; thus, saving people with chest discomfort during observation is a crucial and urgent issue. Normally, emergency care (ER) provides unscheduled outpatient services to patients whose condition requires immediate care. When the demand for emergency care exceeds the medical manpower, the clinicians implement the emergency severity index (ESI), a five-level triage scale for prioritization (where 1 indicates most acute and 5 indicates least acute). Based on the ESI, clinicians can determine which patients should be prioritized swiftly and effectively.

Emotional stress, an indirect indicator associated with CVD, can often manifest through facial expressions. Facial expression is an effective medium for pain communication [5,6,7,8]. To improve pain management quality, Hassan [9] conducted a literature study focusing on non-communicative patients. Based on their studies, the UNBC McMaster Shoulder Pain dataset is the most used dataset. They found that limited research distinguished pain from negative emotions, and most works on detecting pain presence in single images. Ragab [10] revealed that the YOLO model’s capabilities can improve efficiency and accuracy in medical diagnosis and procedures. However, the YOLO model still faces challenges regarding the object scale sensitivity and computational requirements. Advancing technology and artificial intelligence (AI) bring a new horizon in pain detection through facial recognition. Previous studies used facial expression and combined it with body posture to determine the chest pain that is associated with a heart attack. A convolutional neural networks (CNN) approach was employed, and their proposed model achieved 93.33% accuracy [11]. A CNN framework was also employed in the pain assessment study by Balaji [12]. With the data training and data testing ratio at 80:20, respectively, their model can categorize the images as pain or non-pain with a probability of 58%. Meanwhile, Wang [13] conducted pain identification research that used facial expression in the ER and showed the performance of the prediction model achieved its area under the receiver operating characteristic curves (AUC) at 0.633 for severe pain and its AUC at 0.645 for no pain. Using facial recognition for pain assessment was studied by Swetha, who proposed a deep learning model that classified images based on four pain categories, and the accuracy of the model performance reached 75% [14]. Anu [15] proposed a comparative study in automation pain level detection that assists patient monitoring and alerting systems. From the literature study, they stated that most of the proposed frameworks achieved an accuracy of more than 75%. A challenging task for pain recognition through facial expression was experienced by Bargshady, where their joint deep neural network model achieved 75.2% accuracy [16]. Lucey [17] used the active appearance model (AAM) using facial videos of participants suffering from shoulder pain based on the Facial Action Coding System (FACS). They gained the overall average accuracy of action unit (AU) detection at 81.8 with the combined similarity normalized shape and canonical normalized appearance. Facial expressions were also utilized in the pain intensity prediction conducted by Aboussalim [18]. They determined four pain levels with the UNBC dataset and their model obtained an accuracy of 89%.

In the machine learning subsection, the You Only Look Once (YOLO) model, as a single-phase object detector, continues to be developed by researchers in real-time detection of various domains. It has been utilized by various recognition systems, obtaining high performance [19]. Generally, images or videos are analyzed, and the useful features of the images are extracted from the pixel information of images or videos.

Therefore, to support the ER’s standard improvement, this study proposes a non-invasive approach to identify the chest pain based on a vision-based system in facilitating the needs of patients experiencing chest pain for high-priority emergency care. Thus, medical professionals will be able to provide the best treatment to reduce the complications of a patient’s heart attack and improve the golden treatment time. The main contributions of this study can be summarized as follows:To develop an automation pain detection system through facial expression for identifying chest pain.To compare the series of YOLO object detection algorithms.

The paper is organized as follows. Section 2 discusses the materials and research methodology, while Section 3 and Section 4 show the experimental results and discussion, respectively. Finally, the conclusion and a discussion of future work are provided in Section 5.

## 2. Materials and Methods

In this section, the architecture of the proposed model is shown in Figure 1 and the object detection model architecture is depicted in Figure 2, respectively. The pseudocode object detection model is described in the Appendix A.

Data were taken from videos of 1000 patients experiencing chest pain symptoms in the emergency department of Taichung Veterans General Hospital, 3-min questionnaires filled in by patients, and the physician’s evaluations, in line with the ethical standard and approval of the institutional review board. Initially, the medical staff selected between patients who experienced symptoms of chest pain and patients who did not experience chest pain. Volunteers were seated about 1 m from the AXIS network camera which was placed in the ER and positioned to allow consistent capture of facial expressions during sessions. The behaviors of patients are recorded during sessions. In each trial, the total time each volunteer took to complete the experiment was 5 min. The video footage, with a resolution 1920 × 1080 pixels, captured the movements of the patients when pressing on the chest and expressing facial emotions during speech. For image analysis, a frame was manually selected from every 10 frames, and these cutting images represent the different states of the patients during medical treatment. The cutting images were organized in the imaging data sample (CPR dataset), which is shown in Figure 3a. Patients responded to a questionnaire consisting of eight questions related to the chest pain condition. The patients’ descriptive statistics showed that male patients dominated, while the age range of patients was 13 years old to 96 years old, with the mean being 55 years old. The average patient’s heart rate was 86.7 bpm and the mean respiration rate was 18.7 breaths/minutes. The mean triage level was 2.7, which indicates moderate urgency. The average score of disposition was 0.8, which reflects a low likelihood of hospital admission or critical intervention.

For the face detection task, selected video frames must be annotated manually. The dataset was classified into a binary classification: patients with typical chest pain (pain) and atypical chest pain (no pain). The distribution number of samples is defined as follows: the pain category is about 1450 samples, and the other category is about 1500 samples, where the distribution dataset is depicted in Figure 3b.

Researchers explored and studied real-time object recognition, as shown in Table 1. From microscopic coronavirus images, scholars combined the deep learning techniques and image processing technologies used as a diagnostic tool for COVID-19 [20]. They used the YOLOv4 as a single-stage anchor with low computational cost approach, and the performance of YOLOv4 model achieved an accuracy of 94%. In the YOLOv4 family, the backbone structure of the original YOLOv4-tiny model was modified with the Inception-ResNet-A block, and the impact of modification was a slight increase in the model’s performance. The performance of the modified YOLOv4-tiny achieved an accuracy of 99.4% [21]. Other researchers have combined the YOLO deep learning model with the improved block matching method and their proposed model improved the localization and tracking of the myocardial wall, with an average peak signal-to-noise ratio value between 26 and 27. The GoogLeNet architecture was the fundamental construction of their YOLO model [22]. Since animals cannot express themselves in words, Vidal [23] used the YOLOv3 framework for automated face detection to analyze pain in mice. Their recall and precision rates achieved excellent values, at 100%, indicating that the YOLO model has a very high performance. Based on the YOLOv3 framework, Shaik [24] proposed the Novel YOLOv3, which reached an accuracy of 86.65% and outperformed the single-stage detector for face mask detection. Ge [25] also conveyed the drawbacks of a single-stage detector in terms of its effectiveness in detecting small objects and imbalance classes. In neonatal health application, Hausmann [26] utilized the YOLOv3, YOLOv5, and YOLOv6. They found that the YOLOv3 was able to detect the neonate face slightly more than YOLOv5 while YOLOv5 model was capable of showing the instances. However, from the experimental results clearly show that YOLOv6 model achived its accuracy at 62.7%, indicating that it improved accuracy more than YOLOv3 and YOLOv5 in the face detectors. Grooby [27] used the neonatal dataset and compared the YOLOv5 model with the YOLOv7 model. During the training process, the models were configured with a threshold of 20% and optimized using the stochastic gradient descent. The data also experienced data augmentation. The performance of the YOLOv7 model was 0.3% higher than the YOLOv5 model.

Since the COVID-19 pandemic, people have gotten used to wearing masks in public places. Therefore, face detection can be improved to include the parameter of wearing a mask or not [28,29,30,31]. Han [29] conducted a study that identified people wearing face masks with four classes: with mask, without mask, mask worn incorrectly, and mask area. Their proposed model, SMD-YOLOv4-tiny, outperformed the YOLOv4-tiny model, gaining a precision score of 83.78% for wearing masks. The research conducted by Liu [28,32] customized the YOLOv5 model using the CIoULoss as their loss function and modified the scaling method to overcome the operation speed. Therefore, their custom YOLOv5 model was faster than the typical YOLOv5. Jiang [30] suggested that to obtain the best results from YOLOv5’s capabilities, the model should be trained with more than 1500 images per class and more than 10,000 instances per class. Based on previous studies, using the machine learning approach as a subdomain of AI for identifying facial recognition with acute myocardial infarction (AMI) still has positive relevance, even though the degree of chest pain has not been proven to be a reliable indicator of the condition thus far. To validate each method’s accuracy and model reliability in this study, a series of benchmark experiments were employed in the trained models, including accuracy, precision, recall, and F1-score.

**Table 1 diagnostics-15-00017-t001:** Some recent studies on YOLO.

Authors	Best Model	Result	Models
Vidal [23]	YOLOv3	Precision and recall 100%	YOLOv3
Shaik [24]	Novel YOLOv3	Accuracy 86.65%	Novel YOLOv3, Single-stage detector
Doniyorjon [21]	Modified YOLOv4-tiny	Accuracy 99.4%	YOLOv4
Ali [20]	YOLOv4	Accuracy 94%	YOLOv4
Rebaza [33]	YOLOv4	Accuracy 65%	YOLOv4-U_Net
Han [29]	SMD-YOLOv4-tiny	mAP 67.01%	YOLOX_s, YOLOv4, EfficientDet-D1, YOLOv4-tiny
Zhao [31]	YOLOv4	Accuracy 93.56%	YOLOv4
Khalili [34]	YOLOv5	Accuracy 96.7%	YOLOv5-Segment anything model (SAM)
Liu [28]	custom YOLOv5	Precision 78.08%	YOLOv4, YOLOv5s, YOLOv5m, custom YOLOv5
Grooby [27]	YOLOv7	mAP 84.8%	YOLOv7, YOLOv5
Jiang [30]	YOLOv5	Precision 80%	YOLOv5
Liu [32]	ADYOLOv5	Precision 66.5%	YOLOv5s, YOLOv7-tiny, YOLOv8n, ADYOLOv5
Hausmann [26]	YOLOv6	Accuracy 62.7%	YOLOv3, YOLOv5, YOLOv6
Ismail [35]	YOLO-CRNN	Accuracy 89.38%	YOLO-CRNN
Munna [36]	YOLOv6m	mAP 98.86%	YOLOv3, YOLOv4, YOLOv6l, YOLOv6m

## 3. Results

YOLOv4, YOLOv5, YOLOv6, and YOLOv7 were employed in this study. Figure 4a,b show that there is no guarantee that the latest version had the best performance on every dataset. Using the UNBC-McMaster Shoulder Pain Expression dataset [17] and several online sources for people wearing masks as dataset1, the performance of the overall YOLO model achieved accuracies below 75%. The performance of YOLOv7 with the CPR dataset was 35% to 55%, indicating the model’s detection result was poor. While the training results of YOLOv4 and YOLOv6 are the best in the CPR dataset, their performance accuracy ranged between 80% and 100%. However, the YOLOv4 model had the longest training time with the CPR dataset. The comparison results are shown in Table 2.

YOLOv5 had the shortest training time among the proposed YOLO versions. Among the YOLOv5 model, the fastest model in detection task was achieved by the YOLOv5n model, at 0.006 s. This contrasts with the YOLOv5l model, which has the longest detection time at 0.031 s because it had the most significant number of operations at 205.5 giga floating-point operations (GFLOPS). From the series of YOLOv5 approaches, the performance of other models is less accurate, and YOLOv5x6, as the largest model, lacks enough parameter space to make comparisons. Table 3 shows the YOLOv5 families’ performance comparison and Table 4 shows the comparison results of the YOLOv5 custom model series. The detection tasks based on the YOLOv5 series custom model both with pain and non-pain are shown in Figure 5a,b.

The YOLOv5 custom model had a positive prediction for all classes with a confidence score of 0.746, whereas the YOLOv7 custom model had a confidence score of 0.597. The YOLOv5 custom model outperformed the YOLOv7 custom model; the confidence score and precision score curves are depicted in Figure 6a and in Figure 6b, respectively.

The YOLOv5 custom model was more reliable than the YOLOv7 custom model at detecting facial expression of patients experiencing chest pain compared to patients without chest pain, especially at a moderate-to-high confidence level. Table 5 shows the YOLOv5x6 custom model achieved the highest accuracy among the YOLOv5 series. Its accuracy reached more than 55% both in pain and no pain detection. The non-chest pain class in the YOLOv7 custom model demonstrates lower precision; the model exhibits more instability at high confidence thresholds and lower overall precision for chest pain detection. Conversely, the YOLOv5 custom model offers better and more stable performance in detecting chest pain.

The loss function metrics are the indicators that show how good a model is at achieving the desired output. Figure 7 shows the box loss during the training process. The *X*-axis indicates the epoch’s value and the *Y*-axis indicates the loss value. The box loss measures how well the model can predict an object’s position (bounding box) in the image. The box loss graph shows the loss value decreases as the number of epochs increases, which means the YOLOv5 custom model is better at predicting an object’s bounding box. From the point of view of objectness loss, as the number of epochs increases, the loss number also decreases, indicating that the YOLOv5 custom model is getting better at detecting whether there is an object. The classification loss graph shows how good the model is at classifying objects, with a consistent decrease in loss indicating an increase in the model’s ability to classify objects. The other benchmark of the YOLOv5 model is precision, where the precision value increases early in training and stabilizes at a value close to 1, meaning that most of the model’s predictions are correct. Recall measures the number of objects detected compared to the number of objects present. The YOLOv5 custom model’s recall shows improvement, indicating that the model is better at detecting all objects in an image. Overall, the YOLOv5 custom approach’s loss for training and validation continues to decrease, indicating the model is learning well. Meanwhile, the precision and recall increase, indicating that the model’s predictions are getting more accurate and its detection is improving. As with the training process, during the validation process, the box loss has a decrement value, indicating good model generalization to previously unseen data. The increase in the mAP value indicates an improvement in object detection performance at thresholds ranging from 0.5 to 0.95. Similar to the benchmark in the YOLOv5 custom method, however, some different details were found in the YOLOv7 custom model, especially on objectness and classification loss, which is shown in Figure 8. The objectness loss during the validation process highlighted the decline in the trend of fluctuations. Meanwhile, the fluctuations experienced in the classification loss indicate that the model had difficulty in classifying the object with validation data. Overall, the YOLOv7 custom model requires further finetuning due to fluctuations.

The batch size of YOLOv4 and YOLOv6 can be a maximum of 64. There are configuration differences between YOLOv4 and YOLOv6. YOLOv4’s maximum training cycle can be adjusted to 80,000; however, this does not apply to YOLOv6, as YOLOv6’s epoch can only be set to 300. To improve the accuracy of YOLOv4, the maximum training period was optimized and was adjusted from 8000 to 80,000, and it was found that the accuracy significantly increased. Based on the optimization result, the mAP curve steadily increases over the training cycle, reaching a plateau around 80,000 batches. This indicates that the model has learned to detect the object. Higher mAP indicates better performance, as the model is learning quickly initially and then refining its understanding of the data. The YOLOv4 model has the potential to be a useful detection tool in the facial expressions of patients experiencing chest pain. However, the optimization process impacts the training time; it increased from 12 h to 6 days, and the average loss value declined from 0.8 to 0.18. Considering that speed is an important factor in detection, the right choice is to use the YOLOv6 model. The comparison loss value result is depicted in Figure 9.

In Figure 9, the *X*-axis represents the number of iterations processed during the training, whereas the *Y*-axis represents the training loss (blue line) and mAP (red line). The loss behavior shows that as training progresses, the model improves its predictions, resulting in a significant loss reduction from 4.0 to near 1.0. Meanwhile, the mAP curve starts at 59%, showing poor initial detection, but improves as the training progresses, reaching around 78% by the end of 8000 iterations. This suggests that the model’s accuracy increases as it is exposed to more data. A decreasing loss is a positive sign that the model is learning and improving during training. The mAP percentage shows the performance of the model in terms of precision, with it reaching with range from 77 to 78%, which is a good result but may still need improvement for better accuracy, depending on the task at hand.

While the training cycle increased to 80,000, the loss curve starts at a relatively high value at the beginning of the training process, and it decreases sharply over time, indicating that the model had been learned and optimized. The current average loss was 0.1814, a very low value, indicating that the model has reached a good level of convergence. Conversely, the mAP curve starts at a lower value and increases over time, stabilizing around 80–82% after about 10,000 iterations. This metric shows how well the model detects and classifies objects correctly. The research on automated pain assessment with YOLO is limited. An accuracy of 27.8% was reported by Othman [37], who employed the Convolutional Neural Network custom. Xu [38] employed extended multi-task learning, resulting in the mean square error 1.28 ± 0.11. A previous study [26] utilized YOLOv6 with the USF-MNPAD-I dataset, and obtained an accuracy of 62.7%. The performance of the proposed approach is better than the other approaches presented in Table 6.

## 4. Discussion

This study is conducted under the following environment: operating systems Windows 10 and Linux Ubuntu 18.04, language programming Python 3, compute unified device architecture (CUDA), which is a GPU acceleration in NVIDIA’s parallel computing platform, CUDA deep neural network (cuDNN), which is a GPU-accelerated library, zlib compression library, CMake compatibility platform, OpenCV library, Tensorflow framework, Keras framework, and Darknet framework. However, Ubuntu’s performance was shown to be superior to Windows’ operating system; hence, Ubuntu was mostly used. The Tensorflow framework trained the deep learning model once Keras preprocessed the data.

Facial pain detection is the task addressed in this study. The YOLO model, as a one-stage detection method, detects and recognizes the position of an object simultaneously. In this study, all YOLO’s parameters are configured to 64 batches, 300 epochs, 0.001 learning rate, and 415 img_size. Some preliminary training on various models was employed to find the best neural layer weights in the model. The pre-training weights are obtained by training on dataset1 based on the YOLO algorithm. The main limitation in automatic pain recognition is the scarcity of chest pain facial expression databases. One of the public databases used for pain detection through facial expression is the UNBC-McMaster Shoulder Pain database [9,18]. Alghamdi utilized the UNBC-McMaster Shoulder Pain database to alert medical personnel when the patient is in pain by activating an alarm [39]. Ten studies out of fifteen used the UNBC-McMaster Shoulder Pain Archive database to validate their model [6].

The images captured by the camera are stored in the CPR dataset and underwent manipulation techniques. The global and local contrast normalization are used to adjust the images’ contrast. The data augmentation techniques that consist of cropping, rotating, reflecting, and mirroring/flipping were applied to enhance the model’s suitability for real-world uses. The data augmentation results are displayed in Figure 10.

After the normalization process, to prevent the inability of facial emotion recognition while patients wear masks, the vital essential points (such as eyebrows and eyes) were extracted from the images, and the image size was also adjusted, which is displayed in Figure 11. The following process labels data into two categories: pain and no pain. After the labeling stage, the dataset is divided into three parts—data training, data validation, and data testing, with the proportion distributed at 70%, 15%, and 15%, respectively. Among them, the training set is used to estimate the model, the verification set is used to determine the network structure or the parameters that control the complexity of the model, and the test set is used to test and select the performance of the best model.

After the models’ best weights are known, the next phase is to retrain the models with the CPR dataset. The pre-trained YOLO algorithm includes the number of categories, category names, and dataset paths. Medical institutions attach great importance to the time factor to reduce the number of patients and casualties during the golden hour. Therefore, when training the YOLO algorithm, YOLOv5 was prioritized because it had the shortest training time. Due to the large number of internal models in YOLOv5, we compared various models to find the best results. YOLOv5n has the fastest training speed but the lowest number of operations and accuracy among the rest. Therefore, when the training time is extended, the number of operations will increase, and the accuracy will also increase simultaneously.

The comparison structure of YOLOv5 is reflected in Table 4 and Table 5, and the detection results of YOLOv5 models are shown in Figure 4a,b. The next training schema used the various YOLOv5 models to train a custom chest pain expression model; however, its accuracy did not exceed 80%. Therefore, the YOLOv4, YOLOv6, and YOLOv7 versions were used for training and analysis. Even though the accuracy of the YOLOv7 model reached 75% with dataset1, it did not achieve more than 60% accuracy with the CPR dataset. As such, custom chest pain expression models (YOLOv4 and YOLOv6) were used to analyze and assess the data. The YOLOv4 model was optimized with an 80,000-batch training cycle. The custom training between chest pain and no pain by optimizing the YOLOv4 model are shown in Figure 12. After spending more than 12 h with the custom YOLOv4 model, the following training was for the custom YOLOv6 model, which spent the most training time at 5 h. In terms of ensuring that patients experiencing chest pain get high-priority emergency care, the model performance was improved by hyperparameter tuning. The parameter num_repeats determines how many times each stage of the network is repeated. In order to increase the model’s capacity to extract features in the model’s backbone, the num_repeats parameter was adjusted, which is shown in the pseudocode object detection model. The values of num_repeats were arranged as 1, 12, 12, 18, and 6, indicating more layers, which can improve feature extraction. At stage 1 as the shallow layer, the 1 convolutional block was repeated. It captured the low-level features such as edges and textures. Following the next stage, 12 convolutional blocks were repeated, where the process of feature extraction was more detailed for lower-level patterns. The patterns and shapes were extracted from the start of stage 3 and above. In stage 3, 12 convolutional blocks were further analyzed for feature refinement at a deeper level. In order to achieve high-level features, deeper processing was performed with 18 convolutional blocks at stage 4. The object parts and categories were identified in the final stage, with 6 convolutional blocks. However, more repetition is linear with the increase in computational cost and model size. The accuracy, precision, recall, and the F1-score were employed as benchmarking tests on the trained model, which are shown in Figure 6a,b, Figure 7, and Figure 8. Figure 13 shows the benchmarking YOLOv6 custom model in a Precision–Recall curve, which shows the model’s ability to correctly identify facial expressions associated with chest pain. A high precision and recall indicate that the model can accurately detect the expressions with minimal false positives and false negatives. Therefore, the YOLOv6 custom model was used as a reference by clinicians to formulate the best treatment plan to reduce the injury suffered by the patient. The detection result with the YOLOv6 model in the real world is shown in Figure 14. This intelligent medical assistance program helps physicians quickly identify whether a patient’s life is in danger. Even though the severity of chest pain has not been a reliable indicator of AMI up to this point, it would still be beneficial if artificial intelligence could be used for facial recognition to quickly identify patients with AMI. However, there are still limitations to current pain scales, the value of clinical experience for clinical practice, and improvement goals for emergency care quality and cost reduction.

## 5. Conclusions and Future Work

The application of AI in the healthcare sector has become a significant business opportunity and has led to a revolution. Based on the aforementioned, prevention of increasing mortality rates due to cardiovascular disease can be achieved by identifying patients with the highest risk of suffering from cardiovascular disease and ensuring that these patients receive appropriate care and counseling. The proposed models utilize real-time identification of chest pain conditions and assist the clinician–patient consultations. The model’s reliability was evaluated and optimized in chest pain recognition to help overcome the problem of delayed response to cardiac arrest cases in ER during peak-time consultation periods. YOLO’s variances were employed, and more recent YOLO versions did not guarantee suitability for the proposed study. The YOLOv5 model focuses on speed rather than accuracy, whereas the YOLOv7 model focuses on efficiency. The YOLOv4 and YOLOv6 models performed better than YOLOv7 in chest pain recognition. The accuracy of YOLOv4 and YOLOv6 reached 80–100%. Even though they obtain similar accuracy values, the training time for YOLOv6 is faster than YOLOv4. By performing this task, the physician can prioritize the best treatment plan, reduce the extent of cardiac damage in patients, and improve the effectiveness of the golden treatment time. For future work, the chest pain recognition system can be equipped in ambulances to make real-time judgments on whether a patient is experiencing chest pain; in doing so, healthcare professionals can immediately contact a physician and implement emergency measures based on their assessment.

## Figures and Tables

**Figure 1 diagnostics-15-00017-f001:**
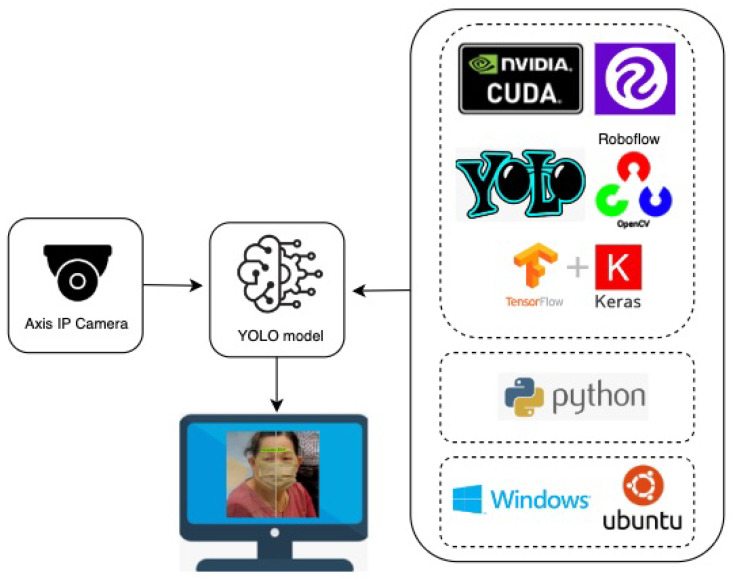
System architecture of the proposed model.

**Figure 2 diagnostics-15-00017-f002:**
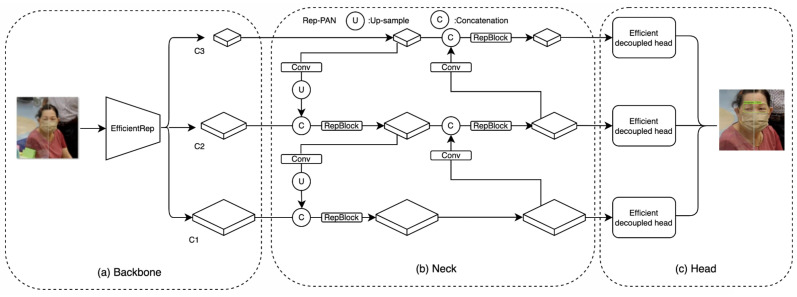
The object detection model architecture.

**Figure 3 diagnostics-15-00017-f003:**
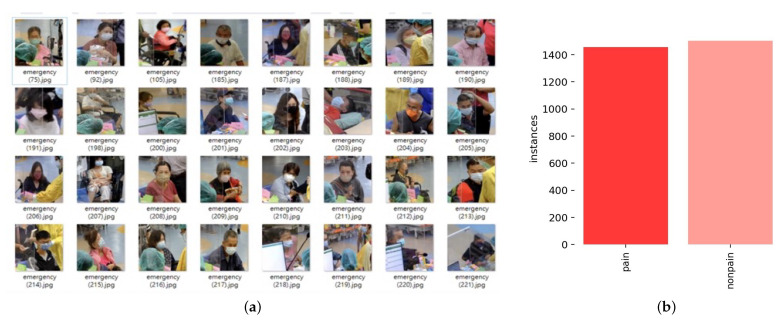
Research dataset. (**a**) Dataset of proposed study. (**b**) Distribution dataset.

**Figure 4 diagnostics-15-00017-f004:**
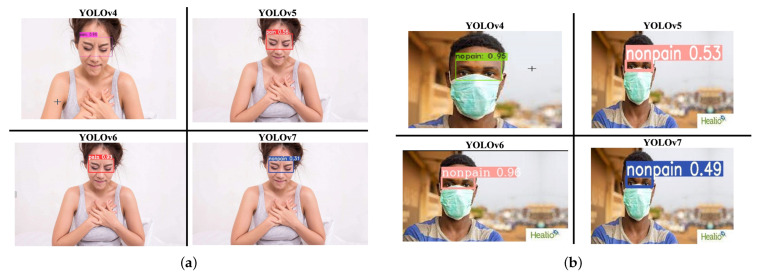
The comparison of YOLO series in chest pain and no chest pain. (**a**) The detection results of YOLO custom series with chest pain. (**b**) The detection results of the YOLO custom series with no chest pain.

**Figure 5 diagnostics-15-00017-f005:**
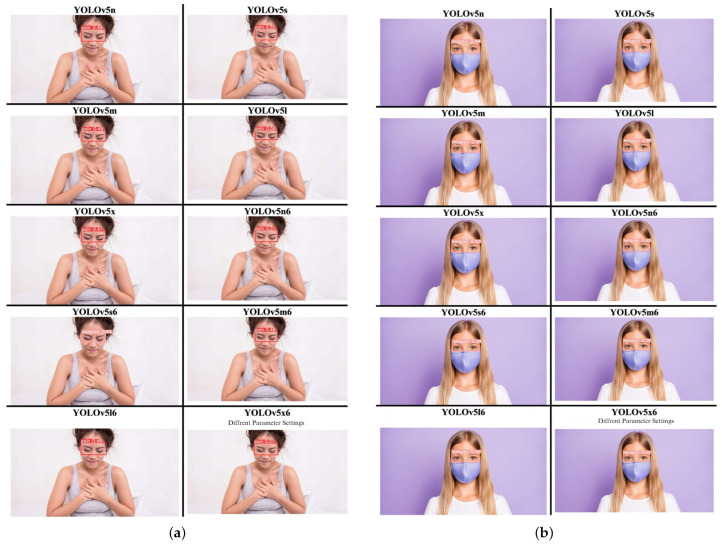
Results of the YOLOv5 series custom model with pain and non-pain. (**a**) YOLOv5 series custom detection results with chest pain. (**b**) YOLOv5 series custom detection results without chest pain.

**Figure 6 diagnostics-15-00017-f006:**
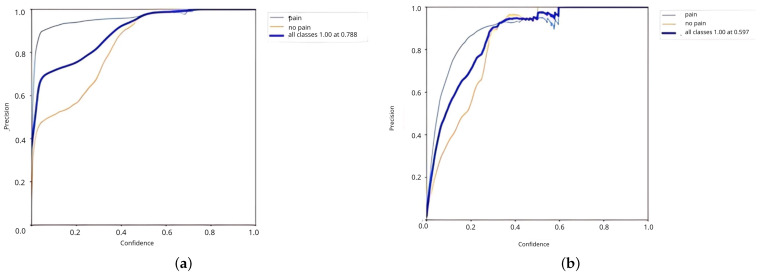
Precision confidence curve of the YOLOv5 and YOLOv7 custom models. (**a**) The precision confidence curve of the YOLOv5 custom model. (**b**) The precision confidence curve of the YOLOv7 custom model.

**Figure 7 diagnostics-15-00017-f007:**
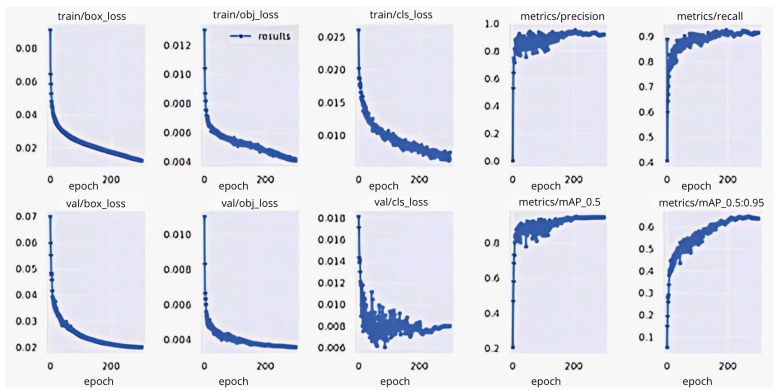
Benchmark metrics of the YOLOv5 custom model.

**Figure 8 diagnostics-15-00017-f008:**
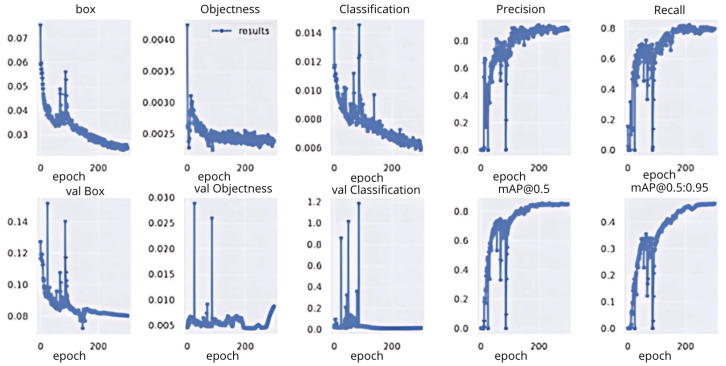
Benchmark metrics of the YOLOv7 custom model.

**Figure 9 diagnostics-15-00017-f009:**
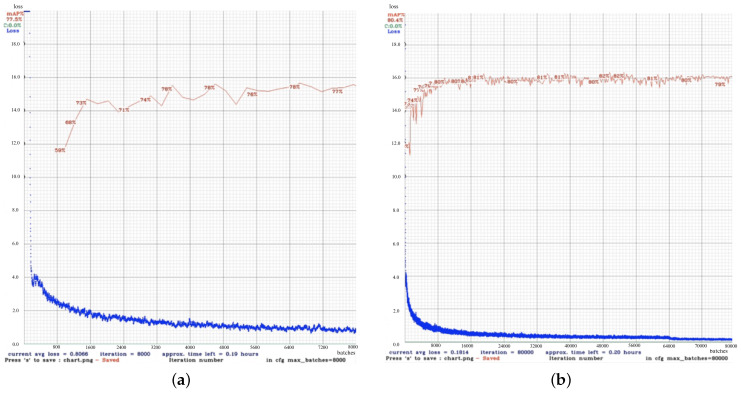
The YOLOv4 custom model results with 8000 batches and 80,000 batches. (**a**) Training cycle with 8000 batches. (**b**) Training cycle with 80,000 batches.

**Figure 10 diagnostics-15-00017-f010:**
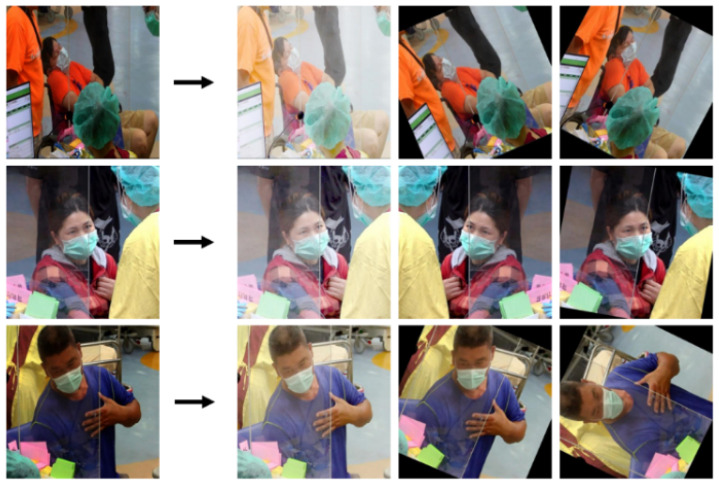
The results of data augmentation.

**Figure 11 diagnostics-15-00017-f011:**
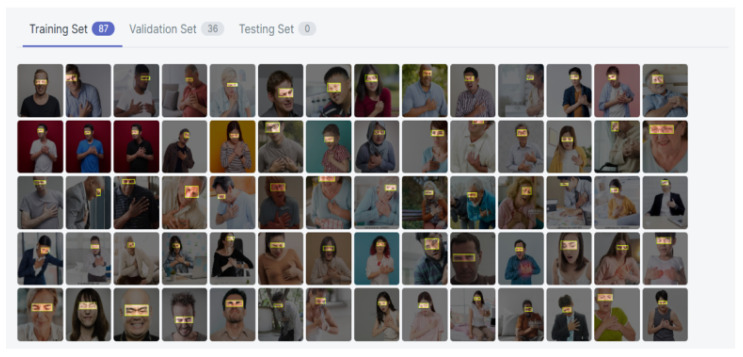
The annotation of vital essential points.

**Figure 12 diagnostics-15-00017-f012:**
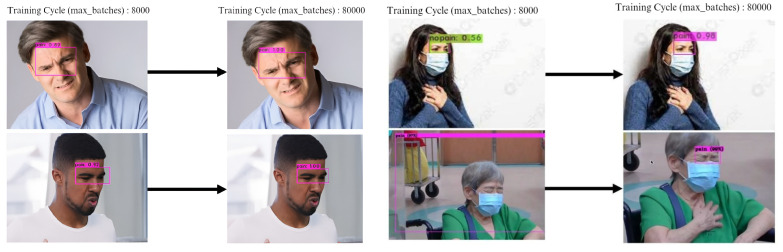
The YOLOv4 custom training pain detection.

**Figure 13 diagnostics-15-00017-f013:**
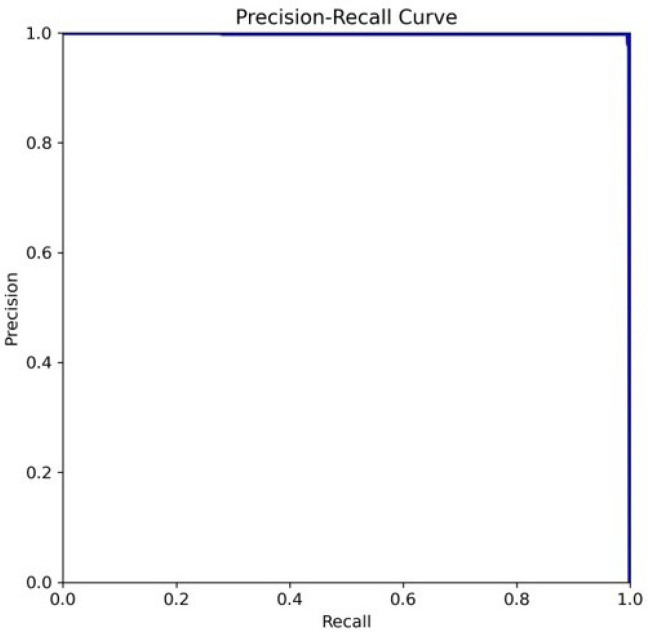
The precision recall curve of the YOLOv6 custom training.

**Figure 14 diagnostics-15-00017-f014:**
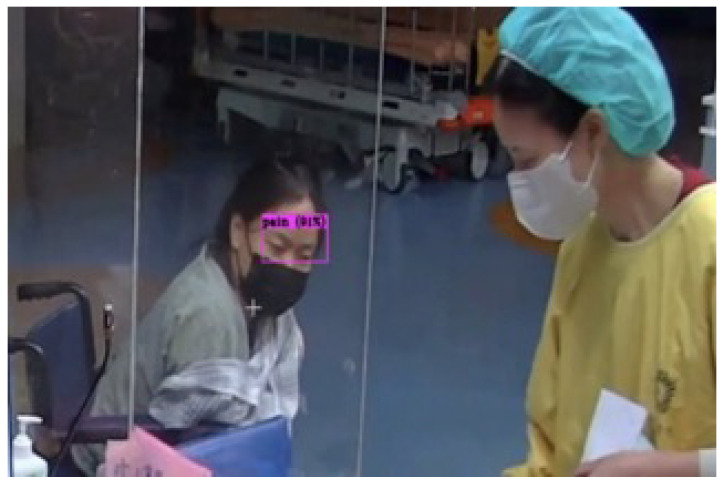
The result of real detection.

**Table 2 diagnostics-15-00017-t002:** Model optimization results.

Model	YOLOv4	YOLOv5	YOLOv6	YOLOv7
Accuracy (dataset1)	50–70%	30–70%	50–70%	50–75%
Accuracy (CPR dataset)	80–100%	50–70%	80–100%	35–55%
Training time (hours)	at least 12	3–4	4–5	5–7

**Table 3 diagnostics-15-00017-t003:** A comparison of the performance of the YOLOv5 families.

Model	mAP	mAP 0.5	Speed CPU (ms)	Speedb1 (ms)	Speed b32	Params (M)	Size (Pixels)
YOLOv5n	28	45.7	45	6.3	0.6	1.9	640
YOLOv5s	37.4	56.8	98	6.4	0.9	7.2	640
YOLOv5m	45.4	64.1	224	8.2	1.7	21.2	640
YOLOv5l	49	67.3	430	10.1	2.7	46.5	640
YOLOv5x	50.7	68.9	766	12.1	4.8	86.7	640
YOLOv5n6	36	54.4	153	8.1	2.1	3.2	1280
YOLOv5s6	44.8	63.7	385	8.2	3.6	12.6	1280
YOLOv5m6	51.3	69.3	887	11.1	6.8	35.7	1280
YOLOv5l6	53.7	71.3	1784	15.8	10.5	76.8	1280
YOLOv5x6	55	72.7	3136	26.2	19.4	140.7	1280

**Table 4 diagnostics-15-00017-t004:** A comparison of YOLOv5 custom models.

Model	Detection Time (s)	FLOP@640 (B)	Training Time (h)	Training Space (MB)
YOLOv5n	0.006	4.50	1.121	3.00
YOLOv5s	0.008	16.4	1.441	14.3
YOLOv5m	0.010	48.9	1.392	42.1
YOLOv5l	0.031	109	3.872	173
YOLOv5x	0.018	205.5	2.496	92.8
YOLOv5n6	0.009	4.60	1.422	6.50
YOLOv5s6	0.008	16.8	1.503	25.0
YOLOv5m6	0.012	49.9	2.147	71.1
YOLOv5l6	0.020	111.2	2.964	153
YOLOv5x6	0.035	209.6	cannot train with the same parameters	cannot train with the same parameters

**Table 5 diagnostics-15-00017-t005:** An accuracy comparison of the structure of the YOLOv5custom model.

Model	Accuracy Pain	Accuracy No Pain
YOLOv5n	0.48	0.44
YOLOv5s	0.50	0.32
YOLOv5m	0.48	0.45
YOLOv5l	0.56	0.60
YOLOv5x	0.47	0.43
YOLOv5n6	0.51	0.40
YOLOv5s6	0.44	0.34
YOLOv5m6	0.48	0.39
YOLOv5l6	0.48	0.42
YOLOv5x6	0.57	0.62

**Table 6 diagnostics-15-00017-t006:** Comparative performance of the proposed model with existing works on facial pain.

Authors	Model	Result
Othman [37]	Convolutional Neural Network custom	Accuracy 27.8%
Xu [38]	Extended Multi-Task Learning	MSE 1.28 ± 0.11
Hausmann [26]	YOLOv6	Accuracy 62.7%
Proposed Model	YOLOv6custom	Accuracy 80–100%

## Data Availability

Dataset is available on request from the authors.

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
