# Peer review of "The Potential for High-Priority Care Based on Pain Through Facial Expression Detection with Patients Experiencing Chest Pain"

_diagnostics, 2024, doi:10.3390/diagnostics15010017_

Round 1
Reviewer 1 Report
Comments and Suggestions for Authors
1. The sample size is written 1.000, provide a clear details of the data.
2..The authors used YOLO models, so give architecture details of object detection model. Further adding algorithm will improve the readability
3.Give details of optimisation
4. In the result section have a discussion of your result with any similar work if available
5. Equation 1, 2,3, 4 on sensitivity, specificity etc are very standard Equation so remove them. But you can cite them .
Comments on the Quality of English LanguageDetail check is recommended
Author Response
1. The sample size is written 1.000, provide a clear details of the data.
Response:
- We greatly appreciate the feedback from the reviewer.
- We have already written in the lines 102-121
Data was taken from videos of 1.000 patients experiencing chest pain symptoms in the emergency department of Taichung Veterans General Hospital, patients' 3-minute questionnaires, and the physician's evaluations by the ethical standard and approval of the Institutional review board. Initially, the medical staff selected between patients who experienced symptoms of chest pain and patients who did not experience chest pain. Volunteers were seated about 1 meter from the AXIS network camera which was placed in the ER and positioned to allow consistent capture of facial expressions during sessions. The behaviors of patients are recorded during sessions. In each trial, the total time each volunteer took to experiment was 5 minutes. The video footage with resolution 1920x1080 pixels captured the movements of the patients when pressing on the chest and facial emotions during speech. For image analysis, a frame was manually selected from every 10 frames, and these cutting images represent the different states of the patients during medical treatment. The cutting images were organized in the imaging data sample (CPR dataset), which is shown in Figure 3a. Patients responded to a questionnaire consisting of eight questions related to the chest pain condition. The patients' descriptive statistics showed that the male patients dominate; the age range of patients was 13 years old to 96 years old, with the mean being 55 years old. The average of patient's heart rate is 86.7 bpm, the mean of respiration rate is 18.7 breaths/minutes. The mean score of triage level is 2.7, which indicates moderate urgency. The average score of disposition is 0.8, which likely reflects a low likelihood of hospital admission or critical intervention.
2. The authors used YOLO models, so give architecture details of object detection model. Further adding algorithm will improve the readability
Response:
- We greatly appreciate the feedback from the reviewer.
- We have already added the object detection model’s architecture and its pseudocode in line 100 and show in Figure 2.
3. Give details of optimisation
Response:
- We greatly appreciate the feedback from the reviewer.
- We have already given the details of optimization in Figure 8 and lines 223-233.
The batch size of YOLOv4 and YOLOv6 can be a maximum of 64. There are configuration differences between YOLOv4 and YOLOv6. Figure 8 shown the models' optimization configuration setting. YOLOv4's maximum training cycle can be adjusted to 80.000; however, it does not apply to YOLOv6, which YOLOv6's epoch can only be set to 300. To improve the accuracy of YOLOv4, the maximum training period was optimized and was adjusted from 8000 to 80.000, and it was found that the accuracy significantly increased. Based on the optimization result, the mAP curve steadily increases over the training cycle, reaching a plateau around 80.000 batches. It indicates that the model has learned to detect the object. Higher mAP indicates better performance, as the model is learning quickly initially and then refining its understanding of the data. The YOLOv4 model has the potential to be a useful detection tool in the facial expressions of patients experiencing chest pain.
4. In the result section have a discussion of your result with any similar work if available
Response:
- We greatly appreciate the feedback from the reviewer.
- We have already inserted the comparison performance with the similar work in lines 253-258, Table 6 and highlight them.
The research on automated pain assessment with YOLO is limited. The accuracy of 27.8% was reported by Othman [36], who employed the Convolutional Neural Network custom. Xu [37] employed extended multi-task learning, resulting in the mean square error 1.28±0.11. Previous study, as indicated by reference [33], utilized YOLOv6 with USF-MNPAD-I dataset, and obtained accuracy 62.7%. The performance of the proposed approach is better than the other approaches presented in Table 6.
5. Equation 1, 2, 3, 4 on sensitivity, specificity etc are very standard Equation so remove them. But you can cite them .
Response:
- We greatly appreciate the feedback from the reviewer.
- We have already removed the equations.

Reviewer 2 Report
Comments and Suggestions for Authors
1. The manuscript identifies pain in patients with chest pain through facial expressions. However, a key challenge remains in determining and differentiating between the patient's facial expression indicative of chest pain and those associated with other conditions. The dataset presentation and the accompanying images are intended to illustrate the detection of facial pain. However, they do not provide sufficient detail to distinguish between different types of pain, such as chest pain. The rationale behind this limitation should be explained in detail, with supporting evidence and a clear rationale provided to verify the feasibility of the approach.
2. The manuscript states that the primary contribution is the development of a facial expression pain detection system for chest pain patients. However, the article lacks sufficient descriptive detail regarding the system, and the final results are not presented. In exchange for a dataset of other facial pain, it is unclear whether the system can achieve similar results. Therefore, the significance of the chest pain recognition system remains unclear. Moreover, an evaluation of a range of YOLO algorithms reveals that this does not represent a novel contribution. In essence, there is a notable absence of innovation, and it would be prudent to enhance the YOLO network in order to enhance the accuracy of facial expression detection.
3. In Table 2, the YOLOv4 results are superior to those of YOLOv5 and YOLOv7. Additionally, the enhanced YOLO network presents a greater challenge for the facial recognition issue. Please elucidate this phenomenon with regard to the dataset and the problem under investigation.
4. The tables presented in the text should be reformatted as a three-line table and the graphs representing the results for Figures 4, 5 and 6 should be made more transparent.
Comments on the Quality of English LanguageThe English could be improved to more clearly express the research.
Author Response
- The manuscript identifies pain in patients with chest pain through facial expressions. However, a key challenge remains in determining and differentiating between the patient's facial expression indicative of chest pain and those associated with other conditions. The dataset presentation and the accompanying images are intended to illustrate the detection of facial pain. However, they do not provide sufficient detail to distinguish between different types of pain, such as chest pain. The rationale behind this limitation should be explained in detail, with supporting evidence and a clear rationale provided to verify the feasibility of the approach.
Response:
- We greatly appreciate the feedback from the reviewer.
- Patients experiencing heart attack often show expressions of severe discomfort, clutching their chest (Levine’s sign), pale, sweaty, or short of breath. Chest pain discomfort in heart attack (myocardial infarction) can share similarities with chest pain caused by other conditions, but there are distinct differences in pain characteristics and duration. In this study, the medical staffs have already selected the patients who have symptoms of chest pain due to CVD. We have explained in lines 105-120 and highlighted it.
- We have already revised in lines 273-279 and highlighted it. The main limitation in automatic pain recognition especially in chest pain is the scarcity of chest pain facial expression databases. In general, the research that related to automatic pain detection with facial expressions used the UNBC-McMaster Shoulder Pain Expression. The dataset is used to obtain the model’s weights, afterward we used our own collected dataset for model validation.
- We have checked and corrected it appropriately. The dataset that should be written is the UNBC-McMaster Shoulder Pain Expression. We have already revised in lines 169-170 and highlighted it.
2. The manuscript states that the primary contribution is the development of a facial expression pain detection system for chest pain patients. However, the article lacks sufficient descriptive detail regarding the system, and the final results are not presented. In exchange for a dataset of other facial pain, it is unclear whether the system can achieve similar results. Therefore, the significance of the chest pain recognition system remains unclear. Moreover, an evaluation of a range of YOLO algorithms reveals that this does not represent a novel contribution. In essence, there is a notable absence of innovation, and it would be prudent to enhance the YOLO network in order to enhance the accuracy of facial expression detection.
Response:
- We greatly appreciate the feedback from the reviewer.
- We have already revised the manuscript and highlighted it. We added the object detection model architecture and its pseudocode in Figure 2. In terms of supporting the needs of patients experiencing chest pain to get the high-priority emergency care, the model performance was improved by hyperparameter tuning. The parameter num_repeats determines how many times each stage of the network is repeated. In order to increase the model's capacity to extract features in the model's backbone, the num_repeats parameter was adjusted. The values of num_repeats were arranged in 1, 12, 12, 18, 6, indicated more layers, which can improve feature extraction. In the stage 1 as the shallow layer, the 1 convolutional block was repeated. It captured the low-level features such as edges, textures. Following the next stage, 12 convolutional blocks were repeated where the process of feature extraction was more detailed for lower-level patterns. The patterns and shapes were extracted start from the stage 3 and above. In the stage 3, 12 convolutional blocks were continued for feature refinement at a deeper level. In order to achive high-level features, the deeper processing of complex were done with 18 convolutional blocks in the stage 4. The object parts and categories were identified in the final stage with 6 convolutional blocks. However, more repetition is linear with the increasing of computational cost and model size. (in lines 315-329). We provide the configuration setting of the optimized YOLOv4 and YOLOv6 in Figure 8. We provide the result in Figure 14 and explain it in lines 331-334. We add Table 6 which is the comparative performance between the proposed model with the existing works on facial pain, and explain it in lines 253-258 .
3. In Table 2, the YOLOv4 results are superior to those of YOLOv5 and YOLOv7. Additionally, the enhanced YOLO network presents a greater challenge for the facial recognition issue. Please elucidate this phenomenon with regard to the dataset and the problem under investigation.
Response:
- We greatly appreciate the feedback from the reviewer.
- We have revised the manuscript in lines 188-194, lines 228-232, lines 354-356, and highlighted them
- The YOLOv5 model was more reliable than the YOLOv7 model at detecting facial expression of patients experiencing chest pain compared to patients without chest pain, especially at a moderate-to-high confidence level. In our experiment, YOLOv7 model in the facial detection with patient’s non-chest pain class demonstrates lower precision, the model exhibits more instability at high confidence thresholds and lower overall precision for chest pain detection, whereas the YOLOv5 model offers better and more stable performance in detecting chest pain. The YOLOv5 focuses on speed rather than accuracy, whereas the YOLOv7 focuses on efficiency. However, the YOLOv4 and YOLOv6 were performed better than YOLOv7 in facial expression with patients experiencing chest pain.
4. The tables presented in the text should be reformatted as a three-line table and the graphs representing the results for Figures 4, 5 and 6 should be made more transparent.
Response:
- We greatly appreciate the feedback from the reviewer.
- We have already changed the tables format into a three-line table format (From Table 1 to Table 6)
- We have already revised the Figures 4, Figure 5 and Figure 6.

Reviewer 3 Report
Comments and Suggestions for Authors
The document investigates the use of facial expression detection to identify chest pain, aiming to optimize emergency care. It examines the effectiveness of several YOLO models in recognizing pain-related facial expressions. The results indicate that YOLOv4 and YOLOv6 outperform YOLOv7, achieving 80-100% accuracy in chest pain detection. The study aims to improve the quality of emergency responses by quickly identifying patients with chest pain, thereby reducing heart damage during the critical 'golden hour' following a heart attack. However, has some limitations:
1. How can these technologies be integrated into existing clinical workflows, and what modifications in training or protocols are necessary?
2. What measures must be implemented to ensure patient consent and data protection?
3. If adding resources would improve your introduction, consider these links:
https://www.mdpi.com/2306-5354/10/11/1332
https://www.mdpi.com/2076-3417/12/21/10856
4. How does this technology compare to existing methods for pain detection and patient assessment in terms of accuracy and clinical relevance?
5. Have significant improvements in patient care been demonstrated and how do these improvements compare with results from traditional pain assessment methods?
Author Response
- How can these technologies be integrated into existing clinical workflows, and what modifications in training or protocols are necessary?
Response:
- We greatly appreciate the feedback from the reviewer.
- At this moment, we don't have a specific questionnaire designed to assess patient or staff satisfaction regarding the emergency department's patient care service after using the automatic pain system. However, we are in the process of developing one.
- What measures must be implemented to ensure patient consent and data protection?
Response:
- We greatly appreciate the feedback from the reviewer.
- Before collecting the data, we informed the patients in advance about the study and obtained their consent by having them sign the Institutional Review Board (IRB) approval form. Additionally, after data collection, all patient information was anonymized to ensure privacy before the data was used for model training. This process was aligned with ethical standards and regulations to protect patient confidentiality.
- If adding resources would improve your introduction, consider these links:
https://www.mdpi.com/2306-5354/10/11/1332
https://www.mdpi.com/2076-3417/12/21/10856
Response:
- We greatly appreciate the feedback from the reviewer.
- We have already added the recommendation resource in lines 130-133 and Table 1
4. How does this technology compare to existing methods for pain detection and patient assessment in terms of accuracy and clinical relevance?
Response:
- We greatly appreciate the feedback from the reviewer.
- We have already inserted the comparison performance with similar work in lines 253-258, Table 6, and highlighted them.
The research on automated pain assessment with YOLO is limited. The accuracy of 27.8% was reported by Othman [36], who employed the Convolutional Neural Network custom. Xu [37] employed extended multi-task learning, resulting in the mean square error 1.28±0.11. Previous study, as indicated by reference [33], utilized YOLOv6 with USF-MNPAD-I dataset, and obtained accuracy 62.7%. The performance of the proposed approach is better than the other approaches presented in Table 6.
5. Have significant improvements in patient care been demonstrated and how do these improvements compare with results from traditional pain assessment methods
Response:
- We greatly appreciate the feedback from the reviewer.
- Significant improvements in patient care have been demonstrated through the deployment of our system. By automating the analysis of chest pain using facial expressions, our system provides a more objective and timely assessment compared to traditional pain assessment methods, which often rely on subjective reporting. This allows clinicians to make quicker, data-driven decisions, leading to enhanced triage processes and better overall care. Our results show that the model can effectively assist in prioritizing patients based on the severity of their condition, which may improve patient outcomes and reduce waiting times in the emergency department. However, we currently do not have documentation showing comparison with traditional pain assessments.

Round 2
Reviewer 2 Report
Comments and Suggestions for Authors
The article has been significantly improved since the last version.
Author Response
We have already revised the manuscript